# An ethogram of acute pain behaviors in cats based on expert consensus

**Sabrine Marangoni**[1], **Julia Beatty**[2], **Paulo V. Steagall**[1,2]*

**1** Faculté de Médecine Vétérinaire, Département de Sciences Cliniques, Université de Montréal, Saint-Hyacinthe, Québec, Canada, **2** Department of Veterinary Clinical Sciences and Centre for Companion Animal Health and Welfare, Jockey Club College of Veterinary Medicine and Life Sciences, City University of Hong Kong, Hong Kong, China

\* pmortens@cityu.edu.hk

## Abstract

An improved understanding of behaviors reflecting acute pain in cats is a priority for feline welfare. The aim of this study was to create and validate a comprehensive ethogram of acute pain behaviors in cats that can discriminate painful versus non-painful individuals. An inventory of behaviors (ethogram) with their respective descriptors was created based on a literature review of PubMed, Web of Science and CAB Abstracts databases. The ethogram was divided into ten behavior categories that could be evaluated by duration and/or frequency: position in the cage, exploratory behaviors, activity, posture and body position, affective-emotional states, vocalization, playing (with an object), feeding, post-feeding and facial expressions/features. Thirty-six behaviors were analyzed independently by four veterinarians with postgraduate qualifications in feline medicine and/or behavior as (1) not relevant, (2) somewhat relevant, (3) quite relevant or (4) highly relevant and used for content (I-CVI) and face validity. Items with I-CVI scores > 0.67 were included. Twenty-four behaviors were included in the final ethogram. Thirteen items presented full agreement (i.e., I-CVI = 1): positioned in the back of the cage, no attention to surroundings, feigned sleep, grooming, attention to wound, crouched/hunched, abnormal gait, depressed, difficulty grasping food, head shaking, eye squinting, blepharospasm and lowered head position. Seven descriptors were reworded according to expert suggestions. The final ethogram provides a detailed description of acute pain behaviors in cats after content and face validity and can be applied to the characterization of different acute painful conditions in hospitalized cats.

## Introduction

Pain is considered a multi-dimensional and complex sensory and affective-emotional response, impacting patient health and welfare [1,2]. Historically, pain management in cats has been neglected, underestimated and overlooked [1,2], partly because recognition of pain requires understanding the cat's unique behavior, and their species-specific responses to the physical and social environment [3]. The solitary nature of cats and their tendency to mask signs of pain as an adaptive strategy in the hospital setting makes acute pain assessment challenging [3]. In addition, behavioral signs of pain may be unique to the individual animal. Cats

**Data Availability Statement:** All relevant data are within the paper and its Supporting Information files.

**Funding:** The authors received no specific funding for this work.

**Competing interests:** The authors have declared that no competing interests exist.

with similar pain conditions may experience different degrees of pain, likely influenced by their genetics, response to analgesics and stressors experienced earlier in life [2,4]. The experience of pain can also be amplified by confounding factors such as comorbidities, fear, anxiety, stress, and/or frustration in the clinical setting. Therefore, implementing routine and individualized pain assessment is vital to provide adequate care while meeting the emotional and environmental needs of the cat [4,5]. Hence, comprehensive observation of posture, activity, attitude and facial expression, and interaction with the environment/observer are required for pain assessment.

There has been a lot of interest in feline acute pain assessment with the publication of specific guidelines [2,4,6], validated multidimensional composite pain scales [7–9], and the Feline Grimace Scale© (FGS) [10]. Additionally, multi-lingual validation of pain scoring systems [11] has improved objectivity while reducing bias during pain assessment. Other than these tools, detailed descriptors of acute pain behaviors have not been published, since behavioral studies using ethograms are rarely reported [12,13]. An ethogram is a catalog that enables an accurate description of species-specific behaviors [14]. Standardization of objective behavior terminology is important to ensure that readers are familiar with commonly used descriptors of pain behaviors. Furthermore, standardization mitigates inter-observer subjectivity and interpretation of clinical manifestations of pain giving confidence to the veterinary care team that observed behaviors are consistent and reliable [14]. Finally, characterization of pain requires an understanding of the duration and frequency of specific behaviors that differentiate painful from non-painful individuals suffering from specific conditions (e.g. medical, surgical, trauma), and involving different anatomic locations (abdominal, thoracic, orthopedic, dental pain, etc.). For example, little is known about dermatologic, aural, visceral and ocular pain in cats, and clinical signs have been mostly described anecdotally or in review articles and case reports [15–18]. Therefore, an ethogram with a detailed description of general feline acute pain behaviors could help with the characterization of these poorly-understood painful conditions.

The aim of this study was to create and validate a comprehensive ethogram of general acute pain behaviors in cage-hospitalized cats. This was a descriptive study without a set working hypothesis.

## Materials and methods

### Databases and search terms

Three databases (PubMed, CAB abstracts, Web of Science) were searched between January 2nd to 20th 2023. No restriction was placed on publication date or language. The search terms were defined using MeSH (Medical Subject Headings), a controlled vocabulary thesaurus used for indexing articles for PubMed. The following descriptors were included: ("acute pain") AND (behavi* OR "pain scoring system" OR "pain indicator" OR "pain assessment" OR ethogram) AND (cats OR feline).

### Eligibility criteria

Original peer-reviewed studies reporting the development and/or validation of pain scoring instruments for the assessment of acute pain, as well as manuscripts reporting the assessment of one or more measurement properties of these instruments were included. Studies involving pain scales during validation of another instrument, the frequency and duration of acute pain behaviors, or reporting lists of pain-related behaviors (i.e., expert consensus using Delphi methodology, narrative reviews, systematic reviews and guidelines) were also included.

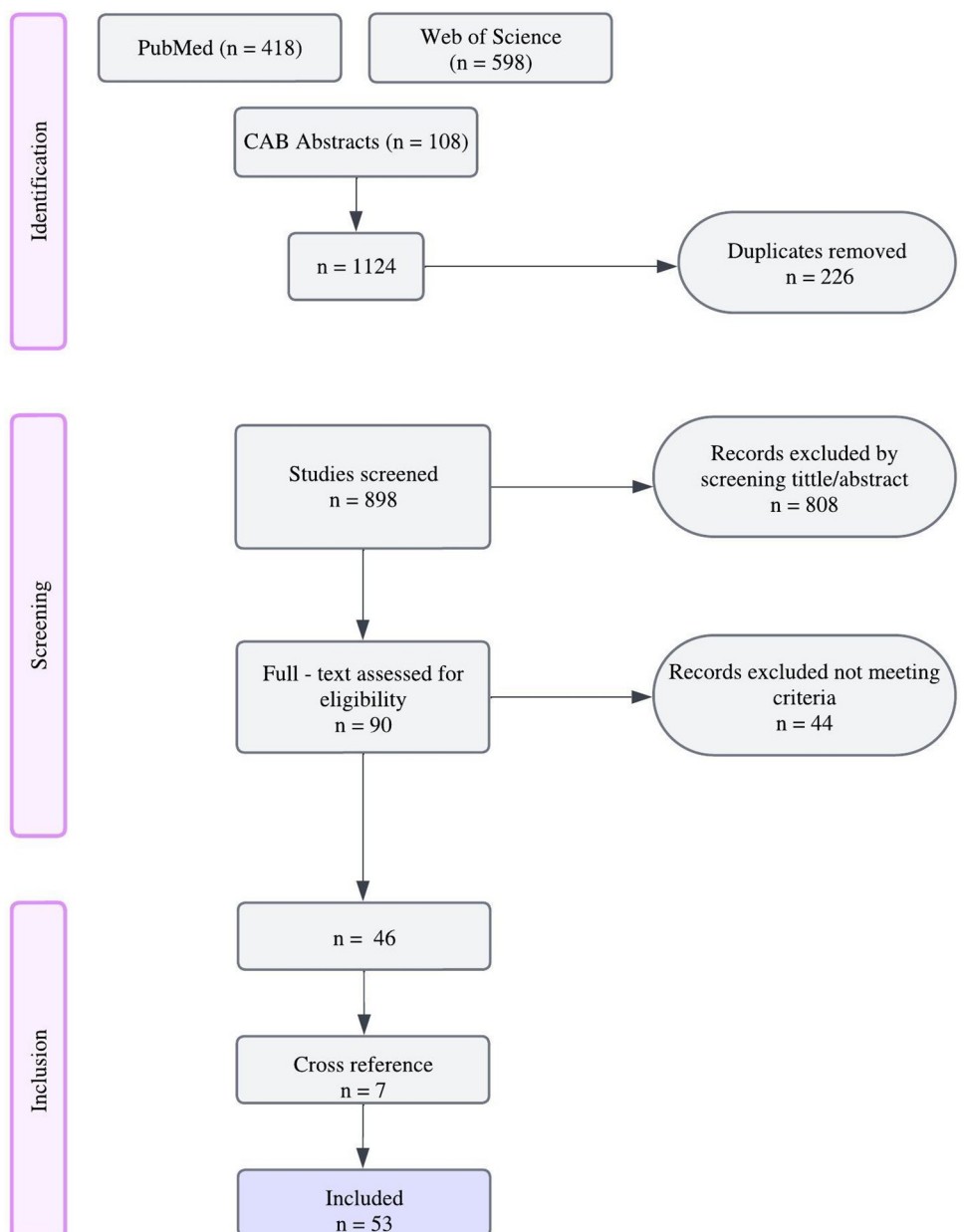

**Fig 1. Flowchart describing the literature review performed for drafting of an ethogram of acute pain behaviors in cats.**

Studies that reported the use of pain scales as an outcome measurement instrument (e.g., in randomized controlled trials comparing two different treatments), nociceptive testing, tools for chronic pain assessment, including objective measures (i.e., activity monitoring) or surveys about knowledge, perceptions and attitudes about pain assessment were excluded.

**Literature search.** Study titles and their abstracts were screened for eligibility by one investigator (SM). Full-text articles were selected, references were exported to Endnote (version 20.0; Clarivate™, Philadelphia) and duplicates were removed. Cross-reference of eligible articles and/or reviews were used to expand the search. A total of 53 articles were included (Fig 1).

## Ethogram

An inventory of behaviors was initially developed by two individuals (SM, PVS). Afterwards, the draft was reviewed by an experienced Royal College of Veterinary Surgeons recognized Specialist in Feline Medicine to increase readability and comprehensiveness of each item (JB). Criteria for inclusion of a behavior in the ethogram were: (1) behaviors that may discriminate painful vs non-painful caged-individuals in the hospital setting to be evaluated in real-time or video assessment including normal behaviors, (2) behaviors that have been described in pain scoring tools, and (3) behaviors that could capture a range of acute pain behaviors in different sources of pain (e.g., trauma, orthopedic, abdominal, dental, ocular, aural pain). The ethogram was divided in categories. Each item within a category included the name of the behavior and a detailed descriptor of the behavior with relevant references.

**Content and face validity.** A panel of four international experienced veterinarians were invited to contribute to this study. Fields of expertise included: a feline specialist board-certified with the American Board of Veterinary Practitioners (ABVP), a board-certified behaviorist from the American College of Veterinary Behaviorists (ACVB), an individual with experience in pain management and animal welfare, and advanced certificate in feline behavior by the International Society of Feline Medicine (Ph.D.; PgDip; ISFMAdvCertFB) and one feline specialist qualified by the Australian and New Zealand College of Veterinary Scientists and advanced certificate in feline behavior (ANZCVS; Ph.D.; PgDip; ISFMAdvCertFB). These individuals were selected based on leadership in feline, behavior and pain medicine. The panel was required to independently analyze each behavioral item of the ethogram as (1) not relevant, (2) somewhat relevant (3) quite relevant or (4) highly relevant for content (I-CVI, item-content validity index) and face validity.

From the formula $CVI = \frac{ne - \frac{N}{2}}{N/2}$ [11], where $ne$ is the number of evaluators who consider the item relevant (scores 3 and 4) and N is the total number of evaluators, items with I-CVI scores > 0.67 [19] were included in the final ethogram. Face validity was defined as whether the descriptors were an adequate reflection of the behavior. The panel evaluated each descriptor for clarity and comprehension and provided feedback on definitions and wording between February 20[th] and March 5[th] 2023 (S1 Table). Subsequently, the authors evaluated the feedback from the panel, and modifications were made accordingly using consensus and I-CVI.

## Results

The ethogram (36 items) submitted for content and face validity with I-CVI scores is shown in Table 1. Following expert review and feedback, 24 behaviors were included in the final ethogram (Table 2).

Twelve behaviors were removed during content validity: positioned in the front of the cage; investigating, scratching, tail twitch, sitting, not pawing but interest/towards the object, not pawing/no interest towards the object, eating, not eating but interest in food, lip licking (post-feeding), head shaking (not related to eating), lip licking (not related to eating). Most behaviors were removed without concern since the antagonistic behavior was present in the ethogram and allowed the differentiation of painful from pain-free cats. A total of thirteen items presented full agreement (i.e., I-CVI = 1): positioned in the back of the cage, no attention to surroundings (exploratory behavior), feigned sleep, grooming and attention to wound (activity), crouched/hunched and abnormal gait (posture & body position), depressed (affective-emotional states), difficulty grasping food, head shaking (feeding), eye squinting, blepharospasm and lowered head position (facial expression or features).

Seven descriptors were reworded according to expert suggestions. The behavior 'withdrawing' was moved from 'exploratory behavior' to the 'activity' category, and reworded as

**Table 1. An ethogram of acute pain behaviors in cats submitted for content and face validity with I-CVI scores.**

| Categories | Behavior | Descriptors | I-CVI |
|---|---|---|---|
| **Position in the cage** | *Back* (D) | Cat is positioned in the back of the cage [9,12] | 1 |
| | *Front* (D) | Cat is positioned in the front of the cage [9,12] | 0.5 |
| **Exploratory behavior** | *Withdrawing* (F) (D) | Move away from a stimulus or an observer [14,20] | 0.75 |
| | *Investigating* (D) | Directed, purposeful movement of the head and/or whole body towards objects or surfaces that are sniffed or contacted with a front paw [14] | 0.5 |
| | *No attention to the surroundings* (D) | Absence of investigatory behaviors [9,13] | 1 |
| **Activity** | *Restlessness* (D) | Unwilling or unable to stay still or to be quiet or calm [7,9,20,21] | 0.75 |
| | *Feigned sleep* (F) (D) | Not related to sleeping. Eyes are partially closed, the ears are pulled apart, and the muzzle is retracted backwards [4,22] | 1 |
| | *Grooming* (D) | Gentle licking, nibbling, biting or chewing of a body part. Includes licking of the medial thoracic limb and wiping it over the body and/or head [14,20] | 1 |
| | *Stretching* (F) | Straighten or extension of body parts making their longer or wider [14] | 0.75 |
| | *Scratching* (F) | Using the extended claws of a back leg to gently scrape the body [14] | 0.25 |
| | *Tail twitch* (F) | A rapid flick of the tail [4,7,14,20] | 0.25 |
| | *Attention to the wound* (F) | Licking or pawing a wound or painful area excessively, or attempting to do so, often moving the head towards the painful area [7-9] | 1 |
| **Posture & Body Position** | *Sitting* (D) | Maintaining the same position with the thoracic limbs extended and the pelvic limbs flexed with the tarsi in contact with the ground [14] | 0.5 |
| | *Crouched/Hunched up* (D) | Leaning forward with the shoulders raised, head down and/or limbs bent [4,8,20-22] | 1 |
| | *Lying dorsoventrally with pelvic limbs extended/contracted* (F) (D) | Lying in dorsoventral recumbency and contracting or extending its pelvic limbs [4,7,9] | 0.75 |
| | *Abnormal gait* (F) | Abnormal gait, but still able to weight bear to some degree on all four limbs [4,20] | 1 |
| | *Non-weight bearing* (F) | Failing to bear weight on a limb [4,20] | 0.75 |
| **Affective-emotional states** | *Depressed* (D) | Unresponsive, immobile [8,12] | 1 |
| **Vocalization**\* | *Repelling* (F) | Any behavior related to a protective response, including biting, attacking, growling, piloerection and tail twitching [1,8] | 0.75 |
| | *Growling* (F) | A low-pitched, throaty, rumbling noise produced while the mouth is closed [8,14] | 0.75 |
| | *Hissing* (F) | A drawn-out, hissing sound produced by rapid expulsion of air from the cats' mouth, usually during exhalation [8,14] | 0.75 |
| **Playing (with an object)**[a] | *Pawing* (D) | Directed contact with the object using the forepaw(s) [14] | 0.75 |
| | *Not pawing but interest to play/towards the object* (D) | Tracking the object with the eye and/or head movement without pawing [13] | 0.25 |
| | *Not pawing/no interest towards the object* (D) | No reaction to or moves away from the object [13] | 0.25 |
| **Feeding**[b] | *Eating* (D) | Approaches food and prehends, chews and swallows normally [14] | 0.25 |
| | *Difficulty grasping or holding food* (F) | Approaches food but unable to properly prehend (grasp or hold food) [13] | 1 |
| | *Head shaking during feeding* (F) | Shakes the head in a side to side movement during eating and/or in the intervals between chewing food with the teeth and swallowing [13] | 1 |
| | *Not eating but interest in food* (D) | Shows an interest by sniffing or licking the food but does not progress the action of eating [9,13] | 0.5 |
| | *Not eating/no interest in food* (D) | No interest to food [9,13] | 0.75 |
| **Post-feeding**[b] | *Lip licking* (F) | Touches the lips and/or nose with tongue [13,14] | 0.5 |
| | *Grooming* (D) | Gentle licking, scratching, nibbling, biting or chewing the fur of a body part. May also include the licking of a front paw and wiping it over the body and/or head [14] | 0.75 |

(*Continued*)

**Table 1.** (Continued)

| Categories | Behavior | Descriptors | I-CVI |
|---|---|---|---|
| **Facial expressions or features** | *Head shaking (not related to eating)* (F) | Episodic shaking of the head in a side to side movement [13] | 0.5 |
| | *Eye squinting* (F) | Continuous orbital tightening [7,9,10,21] | 1 |
| | *Blepharospasm* (F) | Frequent abnormal, uncontrollable contraction/twitching of the eyelid muscle [20] | 1 |
| | *Lip licking (not related to eating)* (F) | Touches the lips and/or nose with tongue [13] | 0.25 |
| | *Lowered head position* (D) | Head held below the shoulder line or angled down with the chin moving towards the chest [10,20] | 1 |

(I-CVI) item content index validity; (D) Duration of behavior; (F) Frequency of behavior.

*when audio is available.

[a] when playing with the cat.

[b] when food is offered.

'withdrawing/hiding' as comments reflected any activity of avoidance including hiding under a blanket or towel, or behind a box, etc. 'Attention to the wound' was modified by replacing the word 'excessively' with 'repeatedly' related to 'licking, pawing, or attempting to lick or paw, a painful area'. The descriptor for 'crouched/hunched up' position included in 'posture & body position' category was modified. Therefore, 'leaning forward with the shoulders raised, head down and/or limbs bent' was changed to 'leaning forward with both pelvic and thoracic limbs flexed and head down'.

The addition of more description of facial expressions or features was suggested by one reviewer for 'feigned sleep'. However, the consensus among the three authors was to maintain a simple description as 'not related to sleeping; eyes are partially closed, the ears are pulled apart, and the muzzle is retracted backwards'.

For the category 'activity', there was a concern about the subjectivity of 'restlessness' by one reviewer. This was originally described as 'unwillingness to stay still or to be quiet or calm'. It was suspected that the words 'quiet and calm' could have different interpretations. However, as in this case, both words are opposite to 'restlessness'; hence, it was decided to maintain the original descriptor, as it should be easy to identify even if the interpretation of calm or quiet may vary.

## Discussion

This study provides an ethogram of acute pain behaviors in cats that has been validated using content and face validity. This ethogram, comprising 24 behaviors divided into 10 categories, represents a tool that can be utilized in subsequent studies to characterize the duration and frequency of behaviors related to manifestations of pain in conditions that are under-studied in cats. The detailed description of behaviors contained in the ethogram supports objective interpretation of common signs of acute pain in cats, particularly for those less familiar with this species. The ethogram may also complement established pain scoring instruments where some behaviors are listed, but not described. Categories in the ethogram are discussed separately below.

### Position in the cage

Being positioned in the 'back' of the cage is commonly reported in painful cats according to guidelines [4], reviews [21–23] and composite pain scoring tools [7,9]. This behavior was observed during a multidisciplinary study for the characterization of oral pain-related behaviors in cats. Cats that have undergone multiple dental extractions due to severe periodontal disease and/or gingivostomatitis spent more time in the back of the cage than cats with minimal periodontal disease [13]. However, this behavior may not be specific to pain if

**Table 2. Ethogram of acute pain behavior in cats following expert consensus.**

| Categories | Behavior | Descriptors |
|---|---|---|
| **Position in the cage** | *Back* (D) | Positioned in the back of the cage [9,12] |
| **Exploratory behavior** | *No attention to the surroundings* (D) | Absence of investigatory behaviors [9,12] |
| **Activity** | *Restlessness* (D) | Unwilling or unable to stay still or to be quiet or calm [7,9,20,24] |
| | *Withdrawing/Hiding* (F) (D) | Moving away from a stimulus or an observer [14,20] |
| | *Feigned sleep* (F) (D) | Not related to sleeping. Eyes are partially closed, the ears are pulled apart, and the muzzle is retracted backwards [4,22] |
| | *Grooming* (D) | Gentle licking, nibbling, biting or chewing of a body part. Includes licking of the medial thoracic limb and wiping it over the body and/or head [14,20] |
| | *Stretching* (F) | Straightening or extending body parts [14] |
| | *Attention to the wound* (F) | Licking or pawing a wound or painful area repeatedly, or attempting to do so, often moving the head towards the painful area [7–9] |
| **Posture & Body position** | *Crouched/Hunched up* (D) | Leaning forward with both pelvic and thoracic limbs flexed and head down [4,8,20–22] |
| | *Lying dorsoventrally with pelvic limbs extended/contracted* (F) (D) | Lying in dorsoventral recumbency and contracting or extending its pelvic limbs [4,7,9] |
| | *Abnormal gait* (F) | Abnormal gait, but still able to bear weight to some degree on all four limbs [4,20] |
| | *Non-weight bearing* (F) | Failing to bear weight on a limb [4,20] |
| **Affective-emotional states** | *Depressed* (D) | Unresponsive, immobile [8,12] |
| | *Repelling* (F) | Any behavior related to a protective response, including biting, attacking, growling, piloerection and tail twitching [1,8] |
| **Vocalization*** | *Growling* (F) | A low-pitched, throaty, rumbling noise produced while the mouth is closed [8,14] |
| | *Hissing* (F) | A drawn-out, hissing sound produced by rapid expulsion of air from the cats' mouth, usually during exhalation [8,14] |
| **Playing (with an object)[a]** | *Pawing* (F) | Directed contact with the object using the forepaw(s) [14] |
| **Feeding[b]** | *Difficulty grasping or holding food* (F) | Approaches food but unable to properly prehend (grasp or hold food) [13] |
| | *Head shaking during feeding* (F) | Shakes the head in a side to side movement during eating and/or in the intervals between chewing food with the teeth and swallowing [13] |
| | *Not eating/no interest in food* (D) | No interest to food [9,13] |
| **Post-feeding[b]** | *Grooming* (D) | Gentle licking, scratching, nibbling, biting or chewing the fur of a body part [14] |
| **Facial expression or features** | *Eye squinting* (F) | Continuous orbital tightening [7,9,10,21] |
| | *Blepharospasm* (F) | Frequent contraction/twitching of the eyelid muscle [20] |
| | *Lowered head position* (D) | Head held below the shoulder line or angled down with the chin moving towards the chest [10,20] |

(D) Duration of behavior; (F) Frequency of behavior.

*when audio is available.

[a]when playing with the cat.

[b]when food is offered.

considered in isolation or during single observation [22,24]. Non-socialized, fearful or shy cats, whether in pain or not, may stay at the back of the cage [1,3]. However, a confident, friendly, and commonly attentive cat that otherwise would seek attention in the front of the cage, may demonstrate changes in behavioral patterns after surgery, trauma or due to a painful medical condition. These cats commonly spend more time in the back of the cage when in pain [4,22].

## Exploratory behavior

Exploratory behaviors, innate for the *felidae* family, are described elsewhere [14]. 'No attention to surroundings' is similarly described as 'inattentive, unresponsive' within the

UNESP-Botucatu multidimensional pain assessment scale (UFEPS) [9] (psychomotor change —comfort) and in its short-version (UFEPS-SF) [7]. The Glasgow Composite Feline Pain Scale (CPMS-F) [12] and its revised version (rCMPS-F) [8] included the item 'unresponsive'. A definition of the behavior as 'absence of investigatory behaviors' helps to reduce subjectivity as the lack of exploring, sniffing, engaging with surroundings or reward-seeking behaviors may equally characterize a painful or an inhibited cat [13]. Unquestionably, avoidance or inhibition of normal behaviors (i.e., inappetence, inactivity, etc.), as protective responses to distress can also be equivocally interpreted as 'no attention to surroundings'. Although the cat may appear unresponsive to the environment and/or observer (i.e., freezing during hospitalization), the motivation may be fear-anxiety-stress, frustration and/or pain or, a mixture of these negative emotions [1,3]. Indeed, cats with severe oral pain were less active (i.e., reduced exploratory behaviors) than cats with minimal periodontal disease after a dental procedure [13].

## Activity

'Feigned sleep' was described in a review article [22] and recently in the ISFM Consensus Guidelines on acute pain management in cats [4]. Frequently, there is a misinterpretation that a cat is asleep after surgery when there is eye squinting, whiskers changes and the head is lower than the body. Indeed, these changes in facial features have been described in painful cats using the FGS [10] and the rCMPS-F [8]. 'Feigned sleep' must be discriminated from normal restful sleep and veterinary health professionals should be educated on these subtle behaviors. During normal sleep, body posture is commonly relaxed and curled up [4,21,24]; cats have been described to adopt a "bagel" or "croissant" position [4]. In a 'feigned sleep' state, cats can appear to be asleep and immobile. However, relaxation behaviors are not observed and cats may be easily disturbed. A shy or fearful cat may use 'feigned sleep' behavior as a passive coping mechanism in stressful environments. The presence of other pain behaviors, such as a crouched/ hunched up position, may accompany changes in facial expressions of a cat in 'feigned sleep', and pain assessment should take into consideration body posture, for example [4,20].

Cats often perform a deep forward stretch when stroked from head to tail, or immediately after waking up [21,22]. However, in painful conditions, 'stretching' behavior is often absent or altered [21,22]. Another intrinsic species-specific behavior and an important indicator of health status is 'grooming'. Postoperative pain is known to alter grooming behavior [13,23–25]. Acute pain or itch are often correlated with localized overgrooming as they have similar neural pathways [26] and excessive grooming could be an attempt to provide relief to pain or itch. Licking, grooming or biting the skin are common indicators of postoperative pain in cats [9,23,25,27–29], dogs [30], horses [31], cattle [32], pigs [33] and rabbits [34]. 'Attention to the wound' was previously reported in the CMPS-F (Question 3), UFEPS (pain expression–miscellaneous behaviors) and UFEPS-SF (Item 2). This behavior was described during video assessment in cats after ovariohysterectomy [28]. Cats were more active and spent less time giving attention to the wound after the administration of butorphanol. However, cats tended to increase their attention to the wound as the effects of butorphanol subsided, potentially corroborating that this behavior is pain-related and relieved by the administration of opioid analgesics. In that study, video observations also demonstrated that painful cats frequently change body positions. Moving from a sitting position, cats would lean forward with flexed thoracic limbs, trunk lowered and head down. At the same time, pelvic limbs would be flexed for a moment, with the cat laying into a crouched position every five to ten minutes [28]. The authors described this behavior as 'restlessness'. This behavior is often present in painful conditions and was described in the UFEPS (psychomotor change–comfort) [9], CMPS-F—Question 2 [12] and the UFEPS-SF–Item 1 [7].

## Posture & body position

Abnormal postures have been reported after abdominal procedures including crouching/hunching, often with the head down [8,9,12,28]. 'Crouched/hunched up' should be considered during pain assessment. However, in the CMPS-F, tensed/crouching position is allocated at Question 2 (score 3) and rigid/hunched position (score 4). Thus, 'crouched/hunched up' may not be considered interchangeably in terms of pain scoring, although it is not clear on what basis the authors of CMPS-F assigned more weight for hunched than crouching, and the difference between these two behaviors remains unclear. A preliminary study described 'crouching' position as 'sternal recumbency, weight bearing on forelimbs' and first reported the body position 'tucked-up appearance' and 'half-tucked-up appearance' to refer to a cat with a 'tensed or partially tensed abdomen in sternal recumbency' [35]. In that study, painful cats spent more time crouching and frequently assumed a 'half-tucked-up appearance'. An expert consensus considered 'hunched up' posture to be relevant and frequent in the presence of pain in cats whereas 'crouching' was not [20]. This demonstrates the importance of terminology standardization and the rationale for the development of this ethogram. Gait abnormalities, such as 'abnormal gait' and 'non-weight bearing' are commonly described in dogs [2] and horses [36] with musculoskeletal acute painful disorders. In cats, these behaviors can be subtle, intermittent, and highly variable, depending on the source and intensity of pain, particularly because non-weight bearing is a form of abnormal gait.

Adopting dorsolateral recumbency while extending and contracting the pelvic limbs are suggestive of abdominal pain. 'Lying dorsoventrally with pelvic limbic extended/contracted' was described in the UFEPS (psychomotor change—posture) [9], UFEPS-SF (Items 1 and 2) [7] and in the ISFM guidelines on the management of acute pain in cats [4]. However, in the authors' experience, this behavior has not been consistently acknowledged in the literature. Consequently, there is a high probability of this behavior being overlooked or under-recognized in the hospital setting, reinforcing the need of appropriate training to perform pain assessment.

## Affective-emotional states

Body movement and postures, and facial expressions are commonly used by cats for emotional and visual communication [1,37]. Specific neurophysiological responses for each type of emotion may be triggered differently within the individual under different contexts, impacting behavioral pain assessment [37]. Nevertheless, individual cats likely experience and express pain in different ways based on previous stressors and affective-emotional states. Protective responses may result in avoidance and/or 'repelling behaviors'. The latter refers to protective responses including swatting, swiping, striking, scratching, biting, piloerection, tail twitching and/or antagonistic vocalizations to increase distance from the observer or stop any interaction. Unfamiliar and stressful events such as transportation to the hospital setting potentially activate and amplify protective responses in the presence of pain [1,38]. Demeanor can have a significant effect on the overall scores of rCMPS-F and the psychomotor but not pain expression section of the UFEPS [39]. A key to mitigate the activation of fear, anxiety or stress, in which mixed emotions can overlap and be misinterpreted, is to provide the cats' environmental demands [5] and reinforce gentle and positive interactions [1]. The latter is important for adequate pain assessment.

## Vocalization

Domestic cats can produce different types of vocalization for inter- or intra-species communication depending on context [3,40,41]. In the case of expressing pain, they can growl, howl,

moan, snarl, spit, yowl or hiss [20]. However, these vocalizations can be altered by the cats' genetics and modulated by the environment and previous experiences [14,41]. During creation of the ethogram, the question was raised whether 'howling' and 'yowling' should be included. However, it was decided that 'growling' and 'hissing' adequately differentiate painful from non-painful cats, as consistently described in previous reports and pain scoring tools [7–9,20]. Additionally, descriptions of vocalization patterns can be subjective since perception is multi-factorial and individually variable, especially for veterinary health professionals whose English is not their first language.

## Playing (with an object)

'Pawing' at objects or other animals may be part of playing behavior in cats. Object play varies between individuals and has a crucial role in motor and cognitive training in early stages of life and socialization [42]. Decreased playfulness can be a behavioral indicator of pain, and has been described in cats with oral pain [13]. Cats in severe pain had decreased duration of 'paw-ing the ribbon' than non-painful cats [13]. Distress in the hospital setting can alter playing motivation. Some cats with outgoing, confident personalities can be overly playful, while shy or fearful individuals may hesitate to play. Individual cats might prefer different types of toys/ribbons, which is subjective as an indicator of pain. Standardizing playing sessions in the hospital setting before and after a painful insult might reduce subjectivity by establishing a baseline. The importance of play behavior is still unclear for pain and welfare assessment and may require further study [42].

## Feeding & post-feeding

Appetite has been previously used in pain assessment in cats [43] and is included in the physiological variables of the UFEPS [9]. Changes in feeding behavior were considered reliable for assessing pain by expert consensus [20], although it requires knowledge of prior feeding patterns. 'Not eating/no interest to food' was considered highly relevant to pain-related behavior in our ethogram and reduced food intake (e.g., after dental surgical procedures) could be of clinical relevance as pain indicator in cats [13,44]. A recent study compared food intake in kittens (i.e., 10 weeks to 6 months of age) undergoing ovariohysterectomy using an opioid-free anesthetic protocol with or without multimodal analgesia. Food intake (%) was significantly higher in non-painful than painful cats when it was evaluated for 2 minutes (15.2 vs 4.2, respectively) and 60 minutes (58.9 vs 29.9, respectively) postoperatively [44]. 'Difficulty grasping or holding food' was observed more frequently in painful when compared to non-painful cats undergoing multiple dental extractions in a multidisciplinary study [13]. In cats, normal behaviors post-feeding include grooming and sleeping [14]. Consequently, the absence of 'grooming' behavior post-feeding is often present in painful conditions.

## Facial expressions or features

Facial expressions of pain have been extensively studied and validated in several species [45]. For example, the Feline Grimace Scale© includes five action units (ear position, orbital tightening, muzzle tension, whisker position, and head position), which are crucial for acute pain assessment. This scale has been robustly validated and is suitable for use by veterinary professionals at different levels of training [46], as well as by caregivers [47]. The behavior 'lowered head posture' was cited in the aforementioned expert consensus [20], whereas 'lowered head position' was described in the FGS [10]. Observation of head position should be included in acute pain assessment.

'Blepharospasm' can be considered an involuntary protective response from a potential injury. It can be present in several ocular conditions, but this behavior is not fully understood or known as an indicator of pain [20]. Painful conditions such as corneal ulcers can trigger both 'blepharospasm' and 'eye squinting', which can be motivated by pain or as an involuntary mechanism to avoid bright light and further injury. A study evaluating the efficacy of topical nalbuphine and oral tramadol for corneal pain used blepharospasm as an important indicator for rescue analgesia [48]. On the other hand, ocular manifestations by viral pathogens are commonly seen in cats, such as feline herpesvirus type 1 (FHV-1) and feline calicivirus (FCV), which can cause conjunctivitis, followed by keratitis, with or without ulceration, especially in cats chronically infected with FHV-1 [49]. Currently, there is a lack of clinical studies reporting if cats with infectious diseases such as FVH-1 or *Chlamydia* frequently present pain-induced blepharospasm.

This study has some limitations. The ethogram requires further validation to confirm that the behaviors described can discriminate painful vs non-painful cats of different life-stages and experiencing different painful conditions. As with many other behaviors, some might be present in conditions that have not been studied. Therefore, some terms that have been removed from the ethogram could still have some clinical relevance. Notably, the ethogram is not a pain scoring system and should not be used for this purpose, and neither to guide clinical decision-making. As a result, it could be used as an additional tool to help veterinary health professionals better recognize the behaviors described. Moreover, the expert panel comprised solely of female veterinarians, who historically provide higher pain scores than men in pain assessment in small animals [50]. Hence, different results may have been obtained with a balanced gender panel or one composed only of male veterinarians. Finally, it is possible that final descriptors and content validity might have been influenced by the size of the panel.

A working knowledge of normal cat behavior (i.e., home environment and hospital setting) is helpful for assessing behavioral signs of pain [4,21]. Additionally, observation and response of the cat after being provided a safe hiding place should facilitate clinical pain assessment [4]. A video-based illustration demonstrating each behavior intended to assist in training veterinary health professionals in acute pain assessment is in progress. This ethogram is now being used to characterize the duration and/or frequency of specific behaviors in kittens undergoing ovariohysterectomy and how behaviors are discriminated between painful and non-painful kittens. In conclusion, this ethogram provides a description of behaviors relevant for acute pain assessment in cats after content and face validity.

## Supporting information

**S1 Table. Supplementary table reporting the individual scoring assignments as determined by the expert panel for item content validity index.**
(DOCX)

**S1 File. Supplementary references of articles included in the literature review.**
(DOCX)

## Acknowledgments

The authors are grateful to the expert panel contributing to this study: Dr. Beatriz P Monteiro, Dr. Diane Frank, Dr. Nathalie Dowgray and Dr. Kelly St-Denis.

## Author Contributions

**Conceptualization:** Paulo V. Steagall.

**Investigation:** Sabrine Marangoni, Paulo V. Steagall.

**Methodology:** Sabrine Marangoni, Julia Beatty, Paulo V. Steagall.

**Project administration:** Paulo V. Steagall.

**Resources:** Sabrine Marangoni, Paulo V. Steagall.

**Supervision:** Paulo V. Steagall.

**Validation:** Sabrine Marangoni, Julia Beatty, Paulo V. Steagall.

**Visualization:** Sabrine Marangoni, Paulo V. Steagall.

**Writing – original draft:** Sabrine Marangoni.

**Writing – review & editing:** Julia Beatty, Paulo V. Steagall.

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
