## [Decision Letter · Decision Letter 0]

21 Aug 2023

PONE-D-23-13666An ethogram of acute pain behaviors in cats based on expert consensusPLOS ONE

Dear Dr. Steagall,

Thank you for submitting your manuscript to PLOS ONE. After careful consideration, we feel that it has merit but does not fully meet PLOS ONE’s publication criteria as it currently stands. Therefore, we invite you to submit a revised version of the manuscript that addresses the points raised during the review process.

ACADEMIC EDITOR: As the main component of the study covers a systematic review of literature, include in the appendix a detailed answers to the Preferred Reporting Items for Systematic Reviews and Meta-Analyses (PRISMA) checklist (http://prisma-statement.org/?AspxAutoDetectCookieSupport=1).

We look forward to receiving your revised manuscript.

Kind regards,

Harvie P. Portugaliza, D.V.M., Ph.D.

Academic Editor

PLOS ONE

Journal Requirements:

Additional Editor Comments:

As the main component of the study covers a systematic review of literature, include in the appendix a detailed answers to the Preferred Reporting Items for Systematic Reviews and Meta-Analyses (PRISMA) checklist (http://prisma-statement.org/?AspxAutoDetectCookieSupport=1).

Reviewers' comments:

Reviewer's Responses to Questions

**Comments to the Author**

1. Is the manuscript technically sound, and do the data support the conclusions?

Reviewer #1: Yes

Reviewer #2: Yes

2. Has the statistical analysis been performed appropriately and rigorously? 

Reviewer #1: N/A

Reviewer #2: I Don't Know

3. Have the authors made all data underlying the findings in their manuscript fully available?

Reviewer #1: Yes

Reviewer #2: Yes

4. Is the manuscript presented in an intelligible fashion and written in standard English?

Reviewer #1: Yes

Reviewer #2: Yes

5. Review Comments to the Author

Reviewer #1: Dear Authors,

Thank you for this interesting study.

Please see my comments here:

Line 45: remove “.” after “and’.

Line 51: is this the Authors’ opinion or evidence from literature? Please rephrase or add references here.

Line 58: I agree of the individual response to pain but I would add a reference here

Line 80-82: you do not address this in your study. I would remove it.

Line 82-83: you describe general acute pain behaviours rather than specific pain behaviour for these conditions. I would remove it or rephrase it.

Line 86 : I would add “general” before “acute “; I would remove the following sentence “that can discriminate painful from non-painful individuals”.

Line 87: Please add here the hypothesis for the content validity.

Line 92: remove ‘’,” after “20th”, similarly line 152 ( also please add space between “wording” and “between”.

Content and face validity: was only one round applied here? Did the panel have the chance to review their collective responses after the first round to ensure a general consensus was reached?

Line 160: table 1: I would change the citation to “an ethogram of acute pain behaviors in cats submitted for content and face validity with I-CVI scores”

Line 175: table 2: I would change the citation to “Ethogram of acute pain behavior in cats following expert consensus”

Line 186: of those descriptors that did not present full agreement, were they re-sent to the panel for further discussion and feedback to reach a consensus?

Line 221: “this” : it is not clear what increases subjectivity. Please rephrase

Line 222: I would either add a reference here regarding misinterpretation by veterinary professional with English not as native language, or remove this sentence. Subjectivity of interpretation is common and can happen to anyone not “especially” to foreigners.

Line 236-238: “A working…signs of pain” . This is an important sentence, but I am wondering if it would be more appropriate in the conclusions. Please remove it from here. Same lines 271-272.

Line 243, 245, 288, 300, etc : “question2, question 3 and question 5. They have not been mentioned before in the main text. Please add or remove from here. Line 243: remove the repeated word “in”

Line 273-278: the aim of this study is to provide an ethogram for acute pain behaviour not a tool for pain assessment. therefore, I would remove this paragraph as not pertinent. Or rephrase it based on the contents.

Line 301: posture&body position: “crouched/hunched up” I am wondering if this term could have been improved/better described based on the fact that there is so much confusion between “crouched” and “hunched”

Reviewer #2: This is a great manuscript, assessing additional behaviors associated with pain and not feeling well in cats. There are a few typos (see attached). The only thing that hasn't been thoroughly addressed are the statistics. This should be addressed in the "Methods".

6. PLOS authors have the option to publish the peer review history of their article (what does this mean?). If published, this will include your full peer review and any attached files.

Reviewer #1: No

Reviewer #2: No

---

## [Author Response · Author response to Decision Letter 0]

25 Aug 2023

Dear Editor,

Comment: As the main component of the study covers a systematic review of literature, include in the appendix a detailed answers to the Preferred Reporting Items for Systematic Reviews and Meta-Analyses (PRISMA) checklist (http://prisma-statement.org/?AspxAutoDetectCookieSupport=1).

Thank you for your comments and for taking the time to act as the handling editor of our manuscript.

The development and validation of this ethogram included a comprehensive literature search and with clear inclusion and exclusion criteria. However, a systematic review was not conducted and, therefore the PRISMA guidelines do not apply. We did not classify studies based on design, or carry out risk or bias assessment, used intervention reporting or an evaluation of quality and strength of evidence. Our laboratory published some systematic reviews with a very different approach than the current ethogram:

https://pubmed.ncbi.nlm.nih.gov/36662813/

https://pubmed.ncbi.nlm.nih.gov/34510132/

https://pubmed.ncbi.nlm.nih.gov/23782347/

Reviewer #1: Dear Authors,

Thank you for this interesting study.

Please see my comments here:

Authors: Dear Reviewers,

Thank you for your valuable input and your willingness to review our study. 

Line 45: remove “.” after “and’.

Authors: Removed

Line 51: is this the Authors’ opinion or evidence from literature? Please rephrase or add references here.

Authors: References have now been added. This is a historical issue in feline medicine and surgery.

Line 58: I agree of the individual response to pain but I would add a reference here

Authors: Reference added.

Line 80-82: you do not address this in your study. I would remove it.

Authors: Thanks for your suggestion. We acknowledge your concern; however, we believe it is important to highlight these literature gaps in the introduction of the manuscript as part of the justification for the study.

Line 82-83: you describe general acute pain behaviours rather than specific pain behaviour for these conditions. I would remove it or rephrase it 

Authors: Added the word ‘general’. 

Line 86 : I would add “general” before “acute “; I would remove the following sentence “that can discriminate painful from non-painful individuals”.

Authors: Thank you. This sentence was amended.

Line 87: Please add here the hypothesis for the content validity.

Authors: Content validity index involves a quantitative evaluation that can result in any degree to which an instrument may be an appropriate sample of descriptors for the construct being measured (i.e. pain). Therefore, it is not possible to set a defined hypothesis giving the quantitative nature of the CVI, which is dependent on expert consensus that could lead to any ‘direction’.

Line 92: remove ‘’,” after “20th”, similarly line 152 ( also please add space between “wording” and “between”.

Authors: Amended.

Content and face validity: was only one round applied here? Did the panel have the chance to review their collective responses after the first round to ensure a general consensus was reached?

Authors: No. Content and face validity was assessed during one round. This approach was guided by the calculation of I-CVI, which was set before panel evaluation and derived from individual and independent assessments of each item. To the authors’ knowledge (Polli & Beck, 2006), a second round of evaluation is not always required in this context, which is different from, for example, the Delphi methodology. 

Line 160: table 1: I would change the citation to “an ethogram of acute pain behaviors in cats submitted for content and face validity with I-CVI scores”

Authors: Done.

-

Line 175: table 2: I would change the citation to “Ethogram of acute pain behavior in cats following expert consensus”

Authors: Done.

Line 186: of those descriptors that did not present full agreement, were they re-sent to the panel for further discussion and feedback to reach a consensus?

Authors: No, they were not as the final I-CVI was predetermined and used as the outcome for expert consensus. 

Line 221: “this” : it is not clear what increases subjectivity. Please rephrase

Authors: The sentence has been deleted as per the comment below.

Line 222: I would either add a reference here regarding misinterpretation by veterinary professional with English not as native language, or remove this sentence. Subjectivity of interpretation is common and can happen to anyone not “especially” to foreigners.

Authors: Sentence deleted as per suggestion.

Line 236-238: “A working…signs of pain” . This is an important sentence, but I am wondering if it would be more appropriate in the conclusions. Please remove it from here. Same lines 271-272.

Authors: Removed and now added to the conclusions (lines 424-427).

Line 243, 245, 288, 300, etc : “question2, question 3 and question 5. They have not been mentioned before in the main text. Please add or remove from here.

Authors: Deleted

Line 243: remove the repeated word “in”

Authors: Deleted

Line 273-278: the aim of this study is to provide an ethogram for acute pain behaviour not a tool for pain assessment. therefore, I would remove this paragraph as not pertinent. Or rephrase it based on the contents.

Authors: Deleted. Please note that references have been reordered.

Line 301: posture&body position: “crouched/hunched up” I am wondering if this term could have 

been improved/better described based on the fact that there is so much confusion between “crouched” and “hunched”

Authors: The distinction between these terms remains somewhat ambiguous. One might argue if there is a true difference or just semantics. These two terms share similarities and are occasionally treated as synonyms. Both of these expressions frequently appear in case reports and reviews, and they occasionally overlap and are even used interchangeably. However, we discussed some of the concerns with the terminology. The definition of "crouching" has been described (Waran et al., 2007), but we could not clearly find one for “hunched up". We believe that mostly important these terms describe posture and body position of cats with pain, which is more important than a precise definition of each. Best wishes,

Reviewer #2: This is a great manuscript, assessing additional behaviors associated with pain and not feeling well in cats. There are a few typos (see attached). The only thing that hasn't been thoroughly addressed are the statistics. This should be addressed in the "Methods".

Authors: Thank you for your valuable input and for taking the time to correct the typos. This is rather unusual but apart from the I-CVI, no other statistical analysis was conducted. We have included a sentence along with the I-CVI equation. Indeed, descriptive studies involving ethograms do not always involve statistical analysis. In regards the use of ‘bread loaf’ position, the authors are not aware of this terminology. Therefore, we could not comment on it or make changes in the manuscript. Best wishes,

---

## [Decision Letter · Decision Letter 1]

18 Sep 2023

An ethogram of acute pain behaviors in cats based on expert consensus

PONE-D-23-13666R1

Dear Dr. Steagall,

We’re pleased to inform you that your manuscript has been judged scientifically suitable for publication and will be formally accepted for publication once it meets all outstanding technical requirements.

Kind regards,

Harvie P. Portugaliza, D.V.M., Ph.D.

Academic Editor

PLOS ONE

Additional Editor Comments (optional):

Reviewers' comments:

Reviewer's Responses to Questions

**Comments to the Author**

1. If the authors have adequately addressed your comments raised in a previous round of review and you feel that this manuscript is now acceptable for publication, you may indicate that here to bypass the “Comments to the Author” section, enter your conflict of interest statement in the “Confidential to Editor” section, and submit your "Accept" recommendation.

Reviewer #1: All comments have been addressed

2. Is the manuscript technically sound, and do the data support the conclusions?

Reviewer #1: Yes

3. Has the statistical analysis been performed appropriately and rigorously? 

Reviewer #1: N/A

4. Have the authors made all data underlying the findings in their manuscript fully available?

Reviewer #1: Yes

5. Is the manuscript presented in an intelligible fashion and written in standard English?

Reviewer #1: Yes

6. Review Comments to the Author

Reviewer #1: Dear Authors,

Thank you for editing the manuscript following the previous comments. It reads well.

7. PLOS authors have the option to publish the peer review history of their article (what does this mean?). If published, this will include your full peer review and any attached files.

Reviewer #1: No

---

## [Editor Report · Acceptance letter]

20 Sep 2023

PONE-D-23-13666R1 

An ethogram of acute pain behaviors in cats based on expert consensus 

Dear Dr. Steagall:

I'm pleased to inform you that your manuscript has been deemed suitable for publication in PLOS ONE. Congratulations! Your manuscript is now with our production department. 

Kind regards, 

on behalf of

Dr. Harvie P. Portugaliza 

Academic Editor

PLOS ONE